# HUMAN-GUIDED COLUMN NETWORKS: AUGMENTING DEEP LEARNING WITH ADVICE

## ABSTRACT

While extremely successful in several applications, especially with low-level representations; sparse, noisy samples and structured domains (with multiple objects and interactions) are some of the open challenges in most deep models. Column Networks, a deep architecture, can succinctly capture such domain structure and interactions, but may still be prone to sub-optimal learning from sparse and noisy samples. Inspired by the success of human-advice guided learning in AI, especially in data-scarce domains, we propose Knowledge-augmented Column Networks that leverage human advice/knowledge for better learning with noisy/sparse samples. Our experiments demonstrate how our approach leads to either superior overall performance or faster convergence.

## 1 INTRODUCTION

The re-emergence of Deep Learning (Goodfellow et al., 2016) has found significant and successful applications in difficult real-world domains such as image (Krizhevsky et al., 2012), audio (Lee et al., 2009) and video processing (Yue-Hei Ng et al., 2015). Deep Learning has also been increasingly applied to structured domains, where the data is represented using *richer symbolic or graph features* to capture relational structure between entities and attributes in the domain. Intuitively, deep learning architectures are naturally suited to learning and reasoning over such multi-relational domains as they are able to capture increasingly complex interactions between features with deeper layers. However, the combinatorial complexity of reasoning over a large number of relations and objects has remained a significant bottleneck to overcome.

Recent work in relational deep learning has sought to address this particular issue. This includes relational neural networks (Kazemi & Poole, 2018; Šourek et al., 2015), relational Restricted Boltzmann machines (Kaur et al., 2017) and neuro-symbolic architectures such as C-ILP (França et al., 2014). In our work, we focus upon the framework of **Column Networks** (CLNs) developed by Pham et al. (2017). Column networks are composed of several (feedforward) mini-columns each of which represents an entity in the domain. Relationships between two entities are modeled through edges between mini-columns. These edges allow for the short-range exchange of information over successive layers of the column network; however, the true power of column networks emerges as the depth of interactions increases, which allows for the natural modeling of long-range interactions.

Column networks are an attractive approach for several reasons: (1) hidden layers of a CLN share parameters, which means that making the network deeper does not introduce more parameters, (2) as the depth increases, the CLN can begin to model feature interactions of considerable complexity, which is especially attractive for relational learning, and (3) learning and inference are linear in the size of the network and the number of relations, which makes CLNs highly efficient. However, like other deep learning approaches, CLNs rely on vast amounts of data and incorporate little to no knowledge about the problem domain. While this may not be an issue for low-level applications such as image or video processing, it is a significant issue in relational domains, since the relational structure encodes rich, semantic information. This suggests that ignoring domain knowledge can considerably hinder generalization.

It is well known that biasing learners is necessary in order to allow them to inductively leap from training instances to true generalization over new instances (Mitchell, 1980). Indeed, the inductive bias towards "simplicity and generality" leads to network architectures with simplifying assumptions through regularization strategies that aim to control the complexity of the neural/deep network.

While deep learning does incorporate one such bias in the form of domain knowledge (for example, through parameter tying or convolution, which exploits neighborhood information), we are motivated to develop systems that can incorporate richer and more general forms of domain knowledge. This is especially germane for deep relational models as they inherently construct and reason over richer representations. Such domain-knowledge-based inductive biases have been applied to a diverse array of machine learning approaches, variously known as advice-based, knowledge-based or human-guided machine learning.

One way in which a human can guide learning is by providing *rules over training examples and features*. The earliest such approaches combined explanation-based learning (EBL-NN, Shavlik & Towell (1989)) or symbolic domain rules with ANNs (KBANN, Towell & Shavlik (1994)). Domain knowledge as *rules over input features* can also be incorporated into support vector machines (SVMs, Cortes & Vapnik (1995); Schölkopf et al. (1998); Fung et al. (2003); Le et al. (2006); Kunapuli et al. (2010b)). Another natural way a human could guide learning is by expressing *preferences* and has been studied extensively within the preference-elicitation framework due to Boutilier et al. (2006). We are inspired by this form of advice as they have been successful within the context of inverse reinforcement learning (Kunapuli et al., 2013), imitation learning (Odom et al., 2015) and planning (Das et al., 2018).

These approaches span diverse machine learning formalisms, and they all exhibit the same remarkable behavior: **better generalization with fewer training examples** because they effectively exploit and incorporate domain knowledge as an inductive bias. This is the prevailing motivation for our approach: to develop a framework that **allows a human to guide deep learning** by incorporating rules and constraints that define the domain and its aspects. Incorporation of prior knowledge into deep learning has begun to receive interest recently, for instance, the recent work on incorporating prior knowledge of color and scene information into deep learning for image classification (Ding et al., 2018). However, in many such approaches, the guidance is not through a human, but rather through a pre-processing algorithm to generate guidance. Our framework is much more general in that a human provides guidance during learning. Furthermore, the human providing the domain advice is not an AI/ML expert but rather a domain expert who provides rules naturally. We exploit the rich representation power of relational methods to capture, represent and incorporate such rules into relational deep learning models.

We make the following contributions: (1) we propose the formalism of Knowledge-augmented Column Networks, (2) we present, inspired by previous work (such as KBANN), an approach to inject generalized domain knowledge in a CLN and develop the learning strategy that exploits this knowledge, and (3) we demonstrate, across four real problems in some of which CLNs have been previously employed, the effectiveness and efficiency of injecting domain knowledge. Specifically, our results across the domains clearly show statistically superior performance with small amounts of data. As far as we are aware, this is the first work on human-guided CLNs.

## 2 BACKGROUND AND RELATED WORK

The idea of several processing layers to learn increasingly complex abstractions of the data was initiated by the perceptron model (Rosenblatt, 1958) and was further strengthened by the advent of the back-propagation algorithm (LeCun et al., 1998). A deep architecture was proposed by Krizhevsky et al. (2012) and have since been adapted for different problems across the entire spectrum of domains, such as, Atari games via deep reinforcement learning (Mnih et al., 2013), sentiment classification (Glorot et al., 2011) and image super-resolution (Dong et al., 2014).

Applying advice to models has been a long explored problem to construct more robust models to noisy data (Fung et al., 2003; Le et al., 2006; Towell & Shavlik, 1994; Kunapuli et al., 2010a; Odom & Natarajan, 2018). For a unified view of variations of knowledge-based neural networks, we refer to Fu (1995). The knowledge-based neural network framework has been applied successfully to various real world problems such as recognizing genes in DNA sequences (Noordewier et al., 1991), microwave design (Wang & Zhang, 1997), robotic control (Handelman et al., 1990) and recently in personalised learning systems (Melesko & Kurilovas, 2018). Combining relational (symbolic) and deep learning methods has recently gained significant research thrust since relational approaches are indispensable in faithful and explainable modeling of implicit domain structure, which is a major limitation in most deep architectures in spite of their success. While extensive literature exists that

aim to combine the two (Sutskever et al., 2009; Rocktäschel et al., 2014; Lodhi, 2013; Battaglia et al., 2016), to the best of our knowledge, there has been little or no work on incorporating the advice in any such framework.

Column networks transform relational structures into a deep architecture in a principled manner and are designed especially for collective classification tasks (Pham et al., 2017). The architecture and formulation of the column network are suited for adapting it to the advice framework. The GraphSAGE algorithm (Hamilton et al., 2017) shares similarities with column networks since both architectures operate by aggregating neighborhood information but differs in the way the aggregation is performed. Graph convolutional networks (Kipf & Welling, 2016) is another architecture that is very similar to the way CLN operates, again differing in the aggregation method. Diligenti et al. (2017) presents a method of incorporating constraints, as a regularization term, which are first order logic statements with fuzzy semantics, in a neural model and can be extended to collective classification problems. While it is similar in spirit to our proposed approach it differs in its representation and problem setup.

## 3 KNOWLEDGE-AUGMENTED COLUMN NETWORKS

Column Networks (Pham et al., 2017) allow for encoding interactions/relations between entities as well as the attributes of such entities in a principled manner without explicit relational feature construction or vector embedding. This is important when dealing with structured domains, especially, in the case of collective classification. This enables us to seamlessly transform a multi-relational knowledge graph into a deep architecture making them one of the robust *relational* deep models. Figure 1 illustrates an example column network, w.r.t. the knowledge graph on the left. Note how each entity forms its own column and relations are captured via the sparse inter-column connectors.

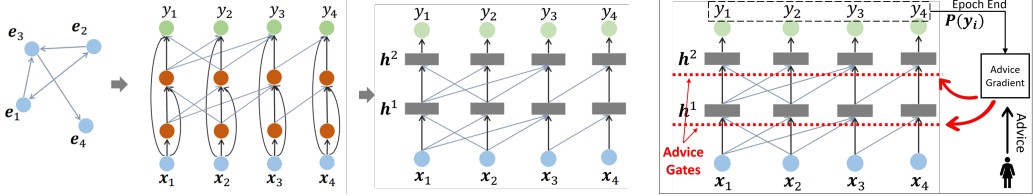

Figure 1: Column network [diag. src.: Pham et al. (2017)]     Figure 2: K-CLN architecture

Consider a graph $\mathcal{G} = (V, A)$, where $V = \{e_i\}_{i=1}^{|V|}$ is the set of vertices/entities. For brevity, we assume only one entity type. However, there is no such theoretical limitation in the formulation. $A$ is the set of arcs/edges between two entities $e_i$ and $e_j$ denoted as $r(e_i, e_j)$. Note that the graph is multi-relational, *i.e.,* $r \in R$ where $R$ is the set of relation types in the domain. To obtain the equivalent Column Network $\mathcal{C}$ from $G$, let $x_i$ be the feature vector representing the attributes of an entity $e_i$ and $y_i$ its label predicted by the model[1]. $h_i^t$ denotes a hidden node w.r.t. entity $e_i$ at the hidden layer $t$ ($t = 1, \ldots, T$ is the index of the hidden layers). As mentioned earlier, the *context* between 2 consecutive layers captures the dependency of the immediate neighborhood (based on arcs/edges/inter-column connectors). Thus, for entity $e_i$, the context w.r.t. $r$ and hidden nodes are computed as,

$$c_{ir}^t = \frac{1}{|\mathcal{N}_r(i)|} \sum_{j \in \mathcal{N}_r(i)} h_j^{t-1} \; ; \qquad h_i^t = g\left(b^t + W^t h_i^{t-1} + \frac{1}{z} \sum_{r \in R} V_r^t c_{ir}^t\right) \qquad (1)$$

where $\mathcal{N}_r(i)$ are all the neighbors of $e_i$ w.r.t. $r$ in the knowledge graph $\mathcal{G}$. Note the absence of context connectors between $h_2^t$ and $h_4^t$ (Figure 1, *right*) since there does not exist any relation between $e_2$ and $e_4$ (Figure 1, *left*). The activation of the hidden nodes is computed as the sum of the bias, the weighted output of the previous hidden layer and the weighted contexts where $W^t \in \mathbb{R}^{K^t \times K^{t1}}$ and $V_r^t \in R^{K^t \times K^{t1}}$ are weight parameters and $b^t$ is a bias for some activation function $g$. $z$ is a pre-defined constant that controls the parameterized contexts from growing too large for complex relations. Setting $z$ to the average number of neighbors of an entity is a reasonable assumption. The

---

[1]Note that since in our formulation every entity is uniquely indexed by $i$, we use $e_i$ and $i$ interchangeably

final output layer is a softmax over the last hidden layer.

$$P(y_i = \ell | h_i^T) = softmax\left(b_l + W_l h_i^T\right) \tag{2}$$

where $\ell \in L$ is the label ($L$ is the set of labels) and $T$ is the index of the last hidden layer.

Following Pham et al. (2017), we choose to formulate our approach in the context of a relation-sensitive predictive modeling, specifically collective classification tasks. However, structured data is implicitly sparse since most entities in the world are not related to each other, thereby adding to the existing challenge of faithful modeling of the underlying structure. The challenge is amplified as *we aim to learn in the presence of knowledge-rich, data-scarce* problems. As we show empirically, sparse samples (or targeted noise) may lead to sub-optimal learning or slower convergence.

**Example 1.** *Consider a problem of classifying whether a published article is about carcinoid metastasis (Zuetenhorst & Taal, 2005) or is irrelevant, from a citation network, and textual features extracted from the articles themselves. There are several challenges: (1) Data is implicitly sparse due to rarity of studied cases and experimental findings, (2) Some articles may cite other articles related to carcinoid metastasis and contain a subset of the textual features, but address another topic and (3) Finally, the presence of targeted noise, where some important citations were not extracted properly by some citation parser and/or the abstracts are not informative enough.*

The above cases may lead to the model not being able to effectively capture certain dependencies, or converge slower, even if they are captured somewhere in the advanced layers of the deep network. Our approach attempts to alleviate this problem via augmented learning of Column Networks using human advice/knowledge. We formally define our problem in the following manner,

---

**Given**: A sparse multi-relational graph $\mathcal{G}$, attribute vectors $x_i$ of each entity (sparse or noisy) in $\mathcal{G}$, equivalent Column-Network $\mathcal{C}$ and access to a Human-expert
**To Do:** More effective and efficient collective classification by knowledge augmented training of $\mathcal{C}$.

---

We develop ***K**nowledge-augmented **C**o**L**umn **N**etworks* (K-CLN), that incorporates human-knowledge, for more effective and efficient learning from relational data (Figure 2 illustrates the overall architecture). While knowledge-based connectionist models are not entirely new, our formulation provides - (1) a principled approach for incorporating advice specified in an intuitive encoding/language based on logic (2) a deep model for collective classification in relational data.

## 3.1 KNOWLEDGE REPRESENTATION

Any model specific encoding of domain knowledge, such as numeric constraints or modified loss functions etc., has several limitations, namely (1) counter-intuitive to the humans since they are domain experts and not experts in machine learning (2) the resulting framework is brittle and not generalizable. Consequently, we employ preference rules (similar to the intuitive IF-THEN statements) to capture human knowledge.

**Definition 1.** *A preference is a modified Horn clause,* $\wedge_{k,x}\mathtt{Attr}_k(\mathtt{E_x}) \wedge \ldots \wedge_{r \in R,x,y} \mathtt{r}(\mathtt{E_x}, \mathtt{E_y}) \Rightarrow [\mathtt{label}(\mathtt{E_z}, \ell_1) \uparrow; \mathtt{label}(\mathtt{E_k}, \ell_2) \downarrow]$, *where* $\ell_1, \ell_2 \in L$ *and the symbols* $\mathtt{E_x}$ *are variables over entities,* $\mathtt{Attr}_k$ *indicate attributes/features of the entity and* $\mathtt{r}$*, a relation.* $\uparrow$ *indicates the preferred label and* $\downarrow$ *indicates the non-preferred ones. Quantification is implicitly* $\forall$ *and hence dropped. We denote a set of preference rules as* $\mathfrak{P}$.

Note that we can always, either have just the preferred label and assume all others as non-preferred, or assume the entire expression as a single literal. Intuitively a rule can be interpreted as, **IF [conditions hold] THEN label** $\ell$ **is preferred**. Note that a preference rule can be partially instantiated as well, *i.e.*, or more of the variables may be substituted with constants.

**Example 2.** *For the prediction task mentioned in Example 1, a possible preference rule could be*

$$\mathtt{hasWord}(\mathtt{E_1}, \text{``A''}) \wedge \mathtt{hasWord}(\mathtt{E_2}, \text{``domain''}) \wedge \mathtt{cites}(\mathtt{E_2}, \mathtt{E_1}) \Rightarrow \mathtt{label}(\mathtt{E_2}, \text{``irrelevant''}) \uparrow$$

*Intuitively, this rule denotes that an article is not a relevant clinical work to carcinoid metastasis if it cites an 'AI' article and contains the word "domains', since it is likely to be another AI article that uses carcinoid metastatis as an evaluation domain.*

### 3.2 Knowledge Injection

Given that knowledge is provided as *partially-instantiated* preference rules $\mathfrak{P}$, more than one entity may satisfy a preference rule. Also, more than one preference rules may be applicable for a single entity. The main intuition is that we aim to consider the error of the trained model w.r.t. both the data and the advice. Consequently, in addition to the *"data gradient"* as with original CLNs, there is a *"advice gradient"*. This gradient acts a feedback to augment the learned weight parameters (both column and context weights) towards the direction of the *advice gradient*. It must be mentioned that not all parameters will be augmented. Only the parameters w.r.t. the entities and relations (contexts) that satisfy $\mathfrak{P}$ should be affected. Let $\mathcal{P}$ be the set of entities and relations that satisfy the set of preference rules $\mathfrak{P}$. The expression for hidden nodes (equation 1) is now modified as,

$$h_i^t = g\left(b^t + W^t h_i^{t-1}\Gamma_i^{(W)} + \frac{1}{z}\sum_{r\in R} V_r^t c_{ir}^t \Gamma_{ir}^{(c)}\right); \text{ where } \Gamma_i, \Gamma_{i,r} = \begin{cases} 1 & \text{if } i,r \notin \mathcal{P} \\ \mathcal{F}(\nabla_i^{\mathfrak{P}}) & \text{if } i,r \in \mathcal{P} \end{cases} \quad (3)$$

where $i \in \mathcal{P}$ and $\Gamma_i^{(W)}$ and $\Gamma_{ir}^{(c)}$ are advice-based soft gates with respect to a hidden node and its context respectively. $\mathcal{F}()$ is some gating function and $\nabla_i^{\mathfrak{P}}$ is the *"advice gradient"*. The key aspect of soft gates is that they attempt to enhance or decrease the contribution of particular edges in the column network aligned with the direction of the *"advice gradient"*. We choose the gating function $\mathcal{F}()$ as an exponential $[\mathcal{F}(\nabla_i^{\mathfrak{P}}) = \exp(\nabla_i^{\mathfrak{P}})]$. The intuition is that soft gates are natural, as they are multiplicative and a positive gradient will result in $\exp(\nabla_i^{\mathfrak{P}}) > 1$ increasing the value/contribution of the respective term, while a negative gradient results in $\exp(\nabla_i^{\mathfrak{P}}) < 1$ pushing them down. We now present the *"advice gradient"* (the gradient with respect to preferred labels).

**Proposition 1.** *Under the assumption that the loss function with respect to advice/preferred labels is a log-likelihood, of the form $\mathcal{L}^{\mathfrak{P}} = \log P(y_i^{(\mathfrak{P})}|h_i^T)$, then the advice gradient is, $\nabla_i^{\mathfrak{P}} = I(y_i^{(\mathfrak{P})}) - P(y_i)$, where $y_i^{(\mathfrak{P})}$ is the preferred label of entity and $i \in \mathcal{P}$ and $I$ is an indicator function over the preferred label. For binary classification, the indicator is inconsequential but for multi-class scenarios it is essential ($I = 1$ for preferred label $\ell$ and $I = 0$ for $L \setminus \ell$).*

Since an entity can satisfy multiple advice rules we take the *MAX* preferred label, *i.e.*, we take the label $y_i^{(\mathcal{P})} = \ell$ to the preferred label if $\ell$ is given by most of the advice rules that $e_j$ satisfies. In case of conflicting advice (i.e. different labels are equally advised), we simply set the advice label to be the label given by the data, $y_i^{(\mathfrak{P})} = y_i$.

***Proof Sketch:*** Most advice based learning methods formulate the effect of advice as a constraint on the parameters or a regularization term on the loss function. We consider a regularization term based on the advice loss $\mathcal{L}^{(\mathfrak{P})} = \log P(y_i = y_i^{(\mathfrak{P})}|h_i^T)$ and we know that $P(y_i|h_i^T) = \text{softmax}(b_\ell + W_\ell h_i^T)$. We consider $b_\ell + W_\ell h_i^T = \Psi_{(y_i,h_i^T)}$ in its functional form following prior non-parametric boosting approaches (Odom et al., 2015). Thus $P(y_i = y_i^{(\mathfrak{P})}|h_i^T) = \exp(\Psi_{(y_i^{(\mathfrak{P})},h_i^T)})/\sum_{y'\in L\setminus y_i^{\mathfrak{P}}}\exp(\Psi_{(y',h_i^T)})$. A functional gradient w.r.t. $\Psi$ of $\mathcal{L}^{(\mathfrak{P})}$ yields

$$\nabla_i^{\mathfrak{P}} = \frac{\partial}{\partial \Psi_{(y_i^{(\mathfrak{P})},h_i^T)}}\log P(y_i = y_i^{(\mathfrak{P})}|h_i^T) = I(y_i^{(\mathfrak{P})}) - P(y_i)$$

Alternatively, assuming a squared loss such as $(y_i^{(\mathfrak{P})} - P(y_i))^2$, would result in an advice gradient of the form $2(y_i^{(\mathfrak{P})} - P(y_i))(1 - P(y_i))P(y_i)$.

As illustrated in the K-CLN architecture (Figure 2), at the end of every epoch of training the *advice gradients* are computed and soft gates are used to augment the value of the hidden units as shown in Equation 3. Algorithm 1 outlines the key steps involved, as described earlier.

## 4 Experiments

We investigate the following questions as part of our experiments, - **(Q1)**: Can K-CLNs learn efficiently with noisy sparse samples? **(Q2)**: Can K-CLNs learn effectively with noisy sparse samples? We compare against the original Column Networks architecture with no advice[2] as a baseline. Our

---

[2]Vanilla CLN refers to the original CLN framework by Pham et al. (2017)

---

**Algorithm 1** K-CLN: **K**nowledge-augmented **Co**Lumn **N**etworks

---

1: **procedure** KCLN(Knowledge graph $\mathcal{G}$, Column network $\mathcal{C}$, Advice $\mathfrak{P}$)
2:     Mask $\mathcal{M}^{\mathcal{P}} \leftarrow$ CREATEMASK($\mathcal{P}$)                    ▷ create mask for all the entities/relations $\in \mathcal{P}$
3:     Initial gradients $\forall i \; \nabla_{i,0}^{\mathfrak{P}} = 0; \; i \in \mathcal{P}$  ▷ Initial gradients set to 0 as no predictions generated at epoch 0
4:     **for** epochs $k = 1$ till convergence **do**                    ▷ convergence criteria same as original CLN
5:         Get gradients $\nabla_{i,(k-1)}^{\mathfrak{P}}$ w.r.t. previous epoch $k-1$
6:         Gates $\Gamma_i^{\mathfrak{P}}, \Gamma_{i,r}^{\mathfrak{P}} \leftarrow \exp\left(\nabla_i^{\mathfrak{P}} \times \mathcal{M}_i^{\mathcal{P}}\right)$
7:         Train using Equation 3
8:         Compute $P(y_i)$ and Store $\nabla_{i,k}^{\mathfrak{P}} \leftarrow I(y_i^{(\mathfrak{P})}) - P(y_i)$          ▷ storing gradients from current epoch
9:     **end for**
10: **end procedure**

---

intention is to show how advice/knowledge can guide model learning towards better predictive performance and efficiency, in the context of collective classification using Column Networks. Hence, we restricted our comparisons to the original work.

**System:** K-CLN has been developed by extending original CLN architecture, which uses *Keras* as the functional deep learning API with a *Theano* backend for tensor manipulation. We extend this system to include: (1) advice gradient feedback at the end of every epoch, (2) modified hidden layer computations and (3) a pre-processing wrapper to parse the advice/preference rules and create appropriate tensor masks. The pre-processing wrapper acts as an interface between the advice encoded in a symbolic language (horn clauses) and the tensor-based computation architecture. The *advice masks* it creates, encode $\mathcal{P}$, *i.e.*, the set of entities and contexts where the gates are applicable.

**Domains:** We evaluate our approach on 4 domains, *Pubmed Diabetes* and *Corporate Messages*, which are multi-class classification problems, and *Internet Social Debates* and *Social Network Disaster Relevance*, which are binary. *Pubmed Diabetes*[3] is a citation network data set for predicting whether a peer-reviewed article is about *Diabetes Type 1, Type 2 or none*, using textual features from pubmed abstracts as well as citation relationships between them. The data set comprises 19717 articles, each of which is considered as an entity with 500 bag-of-words textual features (TF-IDF weighted word vectors), and 44, 338 citation relationships among each other. *Internet Social Debates*[4] is a data set for predicting stance ('for'/'against') about a debate topic from online posts on social debates. It contains 6662 posts (entities) characterized by TF-IDF weighted word vectors (of length 700), extracted from the text and header of the posts, and around 25000 relations, *sameAuthor* and *sameThread*. *Corporate Messages*[5] is an intention prediction data set of 3119 flier messages sent by corporate groups in the finance domain. The target is to predict the intention of the message *(Information, Action or Dialogue)*, using word vectors extracted from text and a network of over $1,000,000$ *sameSourceGroup* relations. Finally, *Social Network Disaster Relevance* (same data source as above) is a relevance prediction data set of 8000 (actual data set is larger but we use a smaller version) *Twitter* posts, curated and annotated by crowd with their relevance to an actual disaster incident. Similar to the the other domains we have 500 bag-of-word features, some confidence score features and $35k$ relations among tweets (of types *'same author'* and *'same location'*).

**Metrics:** Following Pham et al. (2017), we report macro-F1 and micro-F1 scores for the multi-class problems, and F1 scores and AUC-PR for the binary ones. Macro-F1 computes the F1 score independently for each class and takes the average whereas a micro-F1 aggregates the contributions of all classes to compute the average F1 score. For all experiments we use 10 hidden layers and 40 hidden units per column in each layer. All results are averaged over 5 runs. Other settings are consistent with the original CLN framework.

## 4.1 EXPERIMENTAL RESULTS

Recall that our goal is to demonstrate the efficiency and effectiveness of K-CLNs with smaller set of training examples. Hence, we present the aforementioned metrics with varying sample size and with varying epochs and compare our model against *Vanilla CLN*. We split the data sets into a training set and a hold-out test set with 60%-40% ratio. For varying epochs we only learn on 40% of our already split training set (*i.e.*, 24% of the complete data) to train the model with varying epochs and test on the hold-out test set. Figures 3(a) - 3(b) illustrate the micro-F1 and the macro-F1 scores

---

[3]https://linqs.soe.ucsc.edu/data
[4]http://nldslab.soe.ucsc.edu/iac/v2/
[5]https://www.figure-eight.com/data-for-everyone/

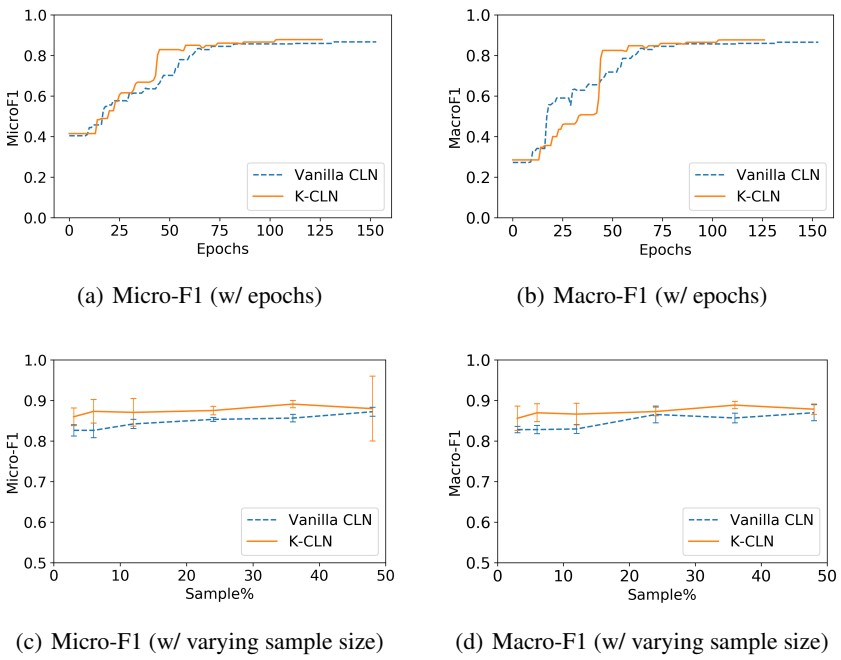

(a) Micro-F1 (w/ epochs)  (b) Macro-F1 (w/ epochs)

(c) Micro-F1 (w/ varying sample size)  (d) Macro-F1 (w/ varying sample size)

Figure 3: **[Pubmed Diabetes publication prediction (multi-class)]** Learning curves - *(Top)* w.r.t. training epochs at 24% (of total) sample, *(Bottom)* w.r.t. varying sample sizes [best viewed in color].

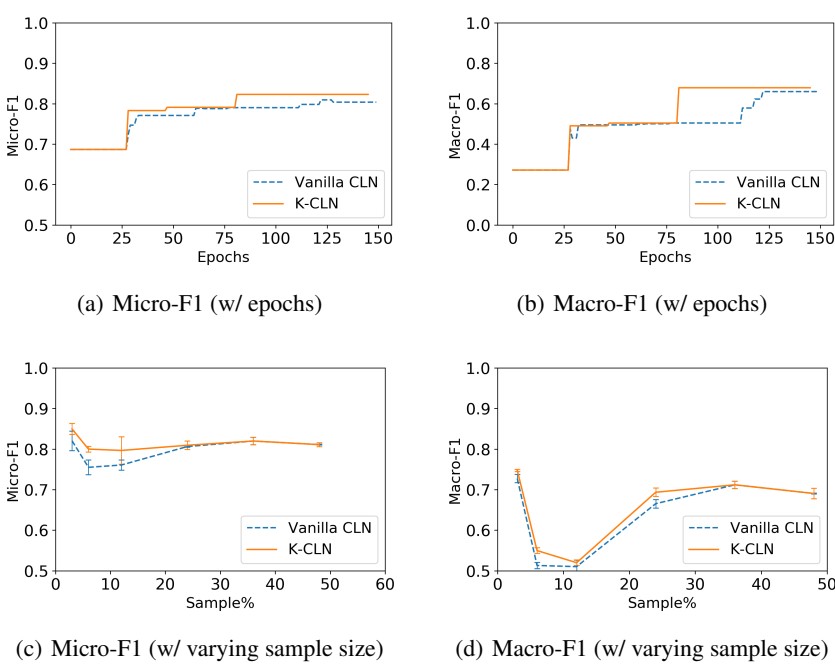

(a) Micro-F1 (w/ epochs)  (b) Macro-F1 (w/ epochs)

(c) Micro-F1 (w/ varying sample size)  (d) Macro-F1 (w/ varying sample size)

Figure 4: **[Corporate Messages intention prediction (multi-class)]** Learning curves - *(Top)* w.r.t. training epochs at 24% (of total) sample, *(Bottom)* w.r.t. varying sample sizes [best viewed in color].

for the *PubMed diabetes* data and Figures 6(a) - 6(b) show the F1 score and AUC-PR for the and social network disaster relevance data. As the figures show, although both K-CLN and Vanilla CLN converge to the same predictive performance, K-CLN converges **significantly faster** (less epochs). For the *corporate messages* and the *internet social debate*, K-CLN not only **converges faster but**

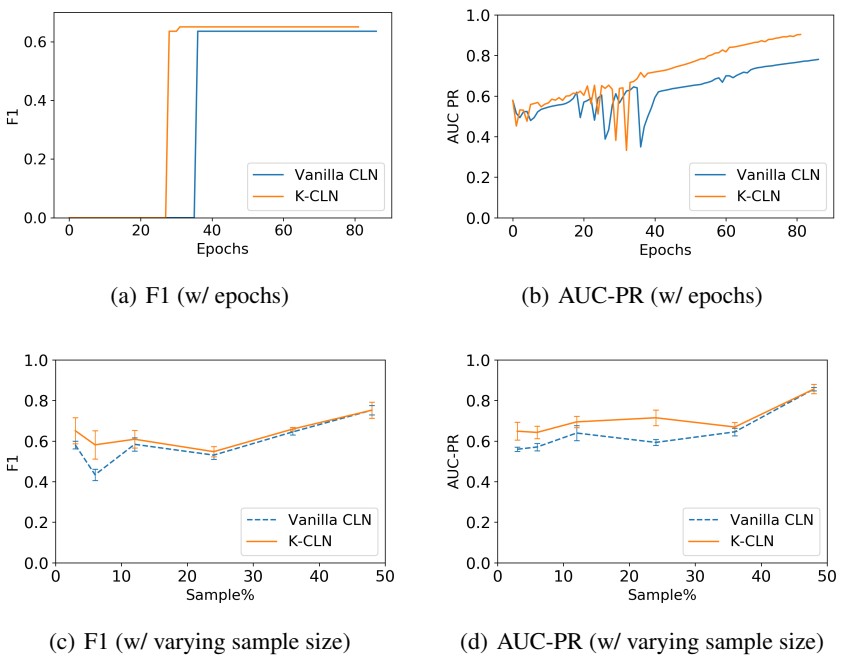

(a) F1 (w/ epochs)       (b) AUC-PR (w/ epochs)

(c) F1 (w/ varying sample size)       (d) AUC-PR (w/ varying sample size)

Figure 5: **[Internet Social debate stance prediction (binary class)]** Learning curves - *(Top)* w.r.t. training epochs at 24% (of total) sample, *(Bottom)* w.r.t. varying sample sizes [best viewed in color].

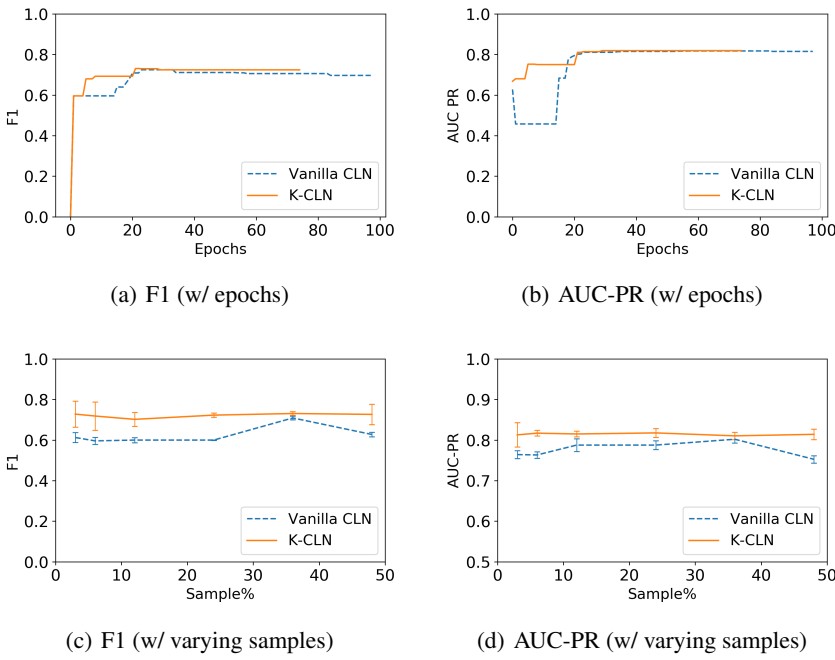

(a) F1 (w/ epochs)       (b) AUC-PR (w/ epochs)

(c) F1 (w/ varying samples)       (d) AUC-PR (w/ varying samples)

Figure 6: **[Social Network Disaster prediction (binary class)]** Learning curves - *(Top)* w.r.t. training epochs at 24% (of total) sample, *(Bottom)* w.r.t. varying sample sizes [best viewed in color].

**also has a better predictive performance** than Vanilla CLN as shown in Figures 4(a) - 4(b) and Figures 5(a) - 5(b). The results show that K-CLNs learn more *efficiently* with noisy sparse samples thereby answering **(Q1)** affirmatively.

Effectiveness of K-CLN is illustrated by its performance with respect to the varying sample sizes of the training set, especially with low sample size. The intuition is, *domain knowledge should help guide the model to learn better when the amount of training data available is small*. K-CLN is trained on gradually varying sample size from 5% of the training data (3% of the complete data) till 80% of the training data (48% of complete data) and tested on the hold-out test set. Figures 3(c) - 3(d) present the micro-F1 and macro-F1 score for *pubMed diabetes* and Figures 6(c) - 6(d) plot the F1 score and AUC-PR for *social network disaster relevance*. It can be seen that K-CLN outperforms Vanilla CLN across all sample sizes, on both metrics, which suggests that the advice is relevant throughout the training phase with varying sample sizes. For *corporate messages*, K-CLN outperforms with small number of samples as shown in the micro-F1 metric (Figure 4(c)) gradually converging to a similar prediction performance with larger samples. Macro-F1 (Figure 4(d)), however, shows that the performance is similar for both the models across all sample sizes, although K-CLN does perform better with very small samples. Since this is a multi-class classification problem, similar performance in the macro-F1 case suggests that in some classes the advice is not applicable during learning, while it applies well w.r.t. other classes, thereby averaging out the final result. For *internet social debate stance prediction*, Figures 5(c) - 5(d) present the F1 score and the AUC-PR respectively. K-CLN outperforms the Vanilla CLN on both metrics and thus we can answer **(Q2)** affirmatively. K-CLNs learn *effectively* with noisy sparse samples.

## 5    CONCLUSION

We considered the problem of providing guidance for CLNs. Specifically, inspired by treating the domain experts as true domain experts and not CLN experts, we developed a formulation based on *preferences*. This formulation allowed for natural specification of guidance. We derived the gradients based on advice and outlined the integration with the original CLN formulation. Our experimental results across different domains clearly demonstrate the effectiveness and efficiency of the approach, specifically in knowledge-rich, data-scarce problems. Exploring other types of advice including feature importances, qualitative constraints, privileged information, etc. is a potential future direction. Scaling our approach to web-scale data is a natural extension. Finally, extending the idea to other deep models remains an interesting direction for future research.

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
