# OpenReview forum: "Human-Guided Column Networks: Augmenting Deep Learning with Advice"
_ICLR.cc/2019/Conference_

### Official Review · AnonReviewer3 · 2018-11-01
**If your human guidance is totally wrong, how your model handle such extreme cases?**

**Rating:** 5
**Confidence:** 5

**Review:**

This paper formulates a new method called human-guided column networks to handle sparse and noisy samples. Their main idea is to introduce human knowledge to guide the previous column network for robust training.

Pros:

1. The authors find a fresh direction for learning with noisy samples. The human advice can be viewed as previledged information.

2. The authors perform numerical experiments to demonstrate the efficacy of their framework. And their experimental result support their previous claims.

Cons:

We have three questions in the following.

1. Motivation: The authors are encouraged to re-write their paper with more motivated storyline. The current version is okay but not very exciting for idea selling. For example, human guidance should be your selling point, and you may not restrict your general method into ColumnNet, which will limit the practical usage.

2. Related works: In deep learning with noisy labels, there are three main directions, including small-loss trick [1], estimating noise transition matrix [2,3], and explicit and implicit regularization [4]. I would appreciate if the authors can survey and compare more baselines in their paper instead of listing some basic ones.

3. Experiment:
3.1 Baselines: For noisy labels, the authors should add MentorNet [1] as a baseline https://github.com/google/mentornet From my own experience, this baseline is very strong.
3.2 Datasets: For datasets, I think the author should first compare their methods on symmetric and aysmmetric noisy data. Besides, the authors are encouraged to conduct 1 NLP dataset.

By the way, if your human guidance is totally wrong, how your model handle such extreme cases? Could you please discuss this important point in your paper?

References:

[1] L. Jiang, Z. Zhou, T. Leung, L. Li, and L. Fei-Fei. Mentornet: Learning data-driven curriculum for very deep neural networks on corrupted labels. In ICML, 2018.

[2] G. Patrini, A. Rozza, A. Menon, R. Nock, and L. Qu. Making deep neural networks robust to label noise: A loss correction approach. In CVPR, 2017.

[3] J. Goldberger and E. Ben-Reuven. Training deep neural-networks using a noise adaptation layer. In ICLR, 2017.

[4] T. Miyato, S. Maeda, M. Koyama, and S. Ishii. Virtual adversarial training: A regularization method for supervised and semi-supervised learning. ICLR, 2016.

---

> ### Author Response · Authors · 2018-11-18
> **Response to review comment #3**
>
> We understand the reviewer's perspective about stronger motivation. However, justification behind our usage of Column networks is as follows - (1) Human advice/knowledge/guidance has been proven to extremely effective in cases of systematic noise in data (Odom et al. 2018). Systematic noise can be attributed to two primary aspects, (i) noise in observation (data recording) and (ii) non-representative sample due to sparsity. (2) The challenges are specifically crucial in structured domains with objects and relations (3) There is limited work w.r.t deep models for modeling in structured/relational noisy domains (4) Column networks have proven to be a successful approach (beating other baselines) for principled modeling in such domains especially for collective classification (5) We show that with noisy sparse samples in structured domains CLNs can perform sub-optimally and human guidance/advice/knowledge can help alleviate that challenge.
>
> We will definitely expand our discussion on quality of advice (in the final version if accepted) highlighting how the model is to be made robust to low quality advice. Note, however, quality of advice is an open challenge in human-in-the-loop research. Also note that average performance of K-CLN never goes substantially below original CLN irrespective of the advice or sample size indicating implicit stability.
>
> We will certainly survey and compare with the literature mentioned here. Thank you for the pointers.

---

### Official Review · AnonReviewer2 · 2018-11-02
**Interesting problem, not well executed idea**

**Rating:** 4
**Confidence:** 4

**Review:**

The  paper  introduces  a  method  to  incorporate  human  advises  to  deep  learning  by  extending  Column  Network  (CLN)  -  a  powerful  graph  neural  network  for  collective  classification.

The  problem  is  quite  interesting  and  is  practical  in  real-world. However, I have some concerns:

Correctness
==========
In the main modification to the CLN in Eq (3), the rule-based gates are introduced to every hidden layer. However, the functional gradient with respect to the "advise gradient" is only computed for the last layer (at the end of Section 3). The exponential gates may cause some instability issue due to its unboundedness.

Evaluation
=========
The  questions  in  experiment  (Can  K-CLNs  learn  efficiently/effectively  with  noisy  sparse  samples?)  do  not  support  the  problem  statement  about  human  advice  incorporation.  Thus,  all  they  did  in  the  experiment  is  trying  to  compete  against  CLN.

I would believe that the improvement (which I trust is real) depends critically on the quality and quantity of the human-crafted rules, much in the same way that feature engineering plays the major roles in the classical structured output prediction. Hence more details about the rules set used in experiments should be given.

Presentation
===========
In  the  experiment  part,  the  authors  need  to  describe  their  model  configuration.  The  presentation  of  datasets  consumes  a  lot  of  space  and  can  be  reduced (e.g., using a table).  This  paper  displays  many  unnecessary  figures  that  consumes  a  lot  of  space.  The  paper  provides  some  unnecessary  text  highlights  in  bold.

---

> ### Author Response · Authors · 2018-11-18
> **Response to review comment #2**
>
> The exponential gate is not theoretically unbounded. The advice gradient itself is bounded between -1 <= \nabla <= +1. So exp(\nabla) is bounded between (0.367, 2.718). However, we do appreciate the reviewer's concern that the effect of the multiplicative gates depend on the original value of the nodes on which the gates are applied. Higher the value of a node more is the effect (if exp(\nabla)>1).  While theoretically this needs validation, in practice, as our experiments show clearly, the effects are indeed quite stable. There has not been any scenario where K-CLN performs worse than original CLN. For a much longer paper, we will certainly explore the theoretical underpinnings.
>
> In existing research on human-guided learning, human advice has been shown to be most effective in case of systematic noise. Such noise can be introduced due to 2 primary reasons (1) Noise in observation (data recording) (2) Noise due to lack of representative samples. Challenge 2 is especially crucial in structured data (objects and interactions), since most interactions in the world are false or unobserved. Our experimental question is directed at these 2 primary challenges. We believe our baseline, original CLN, is justified since they have already been proven to be a superior approach for collective classification in structured data. Our objective is to show that CLNs augmented with richer human inputs than mere labels in noisy sparse samples in structured domains can exhibit superior performance to only label-based learning. We will provide more examples of advice rules used in the experiments in the final version if accepted. Quality of advice is a concern in any human-in-the-loop system. However, reasonably good advice helps alleviate the challenges due to noise and sparsity.
>
> We will shorten the data description. Our objective was to highlight the level of complexity of the different tasks and the impact of sparsity (in context of the dimensionality of the data). The only figures in our work are the original CLN framework and its modifications due to K-CLN. Rest are plots of 2 different types of experiments w.r.t. each data set/domain. While it is possible to condense all data sets into one set of plots, that will immensely hamper the clarity of the presented results. Hence all plots are kept distinct and separate. It will be great if you could guide us to the figures that you deem unnecessary.

---

### Official Review · AnonReviewer1 · 2018-11-06
**A modified column network**

**Rating:** 6
**Confidence:** 3

**Review:**

This work proposes a variant of the column network based on the injection of human guidance. The method does not make major changes to the network structure, but by modifying the calculations in the network. Human knowledge is embodied in a defined rule formula. The method is flexible and different entities correspond to different rules. However, the form of knowledge is limited and simple. Experiments have shown that the convergence speed and results are improved, but not significant.

Minor：
Example 2: "A" -> "AI".

---

> ### Author Response · Authors · 2018-11-18
> **Response to review comment #1**
>
> Preference rules has been proven to be a powerful representation strategy for encoding human knowledge (Odom et al., Frontiers in Robotics and AI, 2018). Note how preference rules encoded in First Order Horn clause logic allows for compactness and generalization over multiple instances and objects. In this work, example advice rules shown are comparatively simple for clarity and brevity. Our formulation has no such limitation. Humans can essentially provide arbitrarily complex clauses (i.e. arbitrary number of literals in the body of the horn clauses).

---

### Meta-Review · Area_Chair1 · 2018-12-17
**Poor results, Evaluating knowledge or method to incorporate it?**

**Confidence:** 4
**Recommendation:** Reject

**Metareview:**

The paper considers the task of incorporating knowledge expressed as rules into column networks. The reviewers acknowledge the need for such techniques, like the flexibility of the proposed approach, and appreciate the improvements to convergence speed and accuracy afforded by the proposed work.

The reviewers and the AC note the following as the primary concerns of the paper:
(1) The primary concerned raised by the reviewers was that the evaluation is focused on whether KCLN can beat one with the knowledge, instead of measuring the efficacy of incorporating the knowledge itself (e.g. by comparing with other forms of incorporating knowledge, or by varying the quality of the rules that were introduced), (2) Even otherwise, the empirical results are not significant, offering slight improvements over the vanilla CLN (reviewer 1), (3) There are concerns that the rule-based gates are introduced but gradients are only computed on the final layer, which might lead to instability, and (4) There are a number of issues in the presentation, where the space is used on redundant information and description of datasets, instead of focusing on the proposed model.

The comments by the authors address some of these concerns, in particular, clarifying that the forms of knowledge/rules are not limited, however, they focused on simple rules in the paper. However, the primary concerns in the evaluation still remain: (1) it seems to focus on comparing against Vanilla-CLN, instead of focusing on the source of the knowledge, or on the efficacy in incorporating it (see earlier work on examples of how to evaluate these), and (2) the results are not considerably better with the proposed work, making the reviewers doubtful about the significance of the proposed work.

The reviewers agree that the paper is not ready for publication.